

# Chemical modeling of the reactivity of short-lived greenhouse gases: a model inter-comparison prescribing a well-measured, remote troposphere

Michael J. Prather[1], Clare M. Flynn[1], Xin Zhu[1], Stephen D. Steenrod[2,3], Sarah A. Strode[2,3], Arlene M. Fiore[4], Gustavo Correa[4], Lee T. Murray[5], Jean-Francois Lamarque[6]

[1]Department of Earth System Science, University of California, Irvine, CA 92697-3100, USA
[2]NASA Goddard Space Flight Center, Greenbelt, MD, USA
[3]Universities Space Research Association (USRA), GESTAR, Columbia, MD, USA
[4]Department of Earth and Environmental Sciences and Lamont-Doherty Earth Observatory of Columbia University, Palisades, NY, USA
[5]Department of Earth and Environmental Sciences, University of Rochester, Rochester, NY 14627-0221, USA
[6]Atmospheric Chemistry, Observations and Modeling Laboratory, National Center for Atmospheric Research,
Boulder, CO 80301, USA

Correspondence to: Michael J. Prather (mprather@uci.edu)

***Abstract.*** We develop a new protocol for merging in situ measurements with 3-D model simulations of atmospheric chemistry with the goal of integrating over the data to identify the most reactive air parcels in terms of tropospheric production and loss of the greenhouse gases ozone and methane. Presupposing that we can accurately measure atmospheric composition, we examine whether models constrained by such measurements agree on the chemical budgets for
ozone and methane. In applying our technique to a synthetic data stream of 14,880 parcels along 180W, we are able to isolate the performance of the photochemical modules operating within their global chemistry-climate and chemistry-transport models, removing the effects of modules controlling tracer transport, emissions, and scavenging. Differences in reactivity across models are driven only by the chemical mechanism and the diurnal cycle of photolysis rates, which are
driven in turn by temperature, water vapor, solar zenith angle, clouds, and possibly aerosols and overhead ozone, which are calculated in each model. We evaluate six global models and identify their differences and similarities in simulating the chemistry through a range of innovative



diagnostics. All models agree that the more highly reactive parcels dominate the chemistry (e.g., the hottest 10% of parcels control 25-30% of the total reactivities), but do not fully agree

on which parcels comprise the top 10%. Distinct differences in specific features occur, including the regions of maximum ozone production and methane loss, as well as in the relationship between photolysis and these reactivities. Unique, possibly aberrant, features are identified for each model, providing a benchmark for photochemical module development. Among the 6 models tested here, 3 are almost indistinguishable based on the inherent variability caused by

clouds, and thus we identify 4, effectively distinct, chemical models. Based on this work, we suggest that water vapor differences in model simulations of past and future atmospheres may be a cause of the different evolution of tropospheric $O_3$ and $CH_4$, and lead to different chemistry-climate feedbacks across the models.

*1. Introduction*

The daily passage of sunlight through the lower atmosphere drives photochemical reactions that control many short-lived greenhouse gases (GHGs) and other pollutants. This daily cycle occurs across a range of different chemical compositions; such that even neighboring air parcels can

exhibit a wide range in their reactivity with respect to GHGs (Prather et al., 2017; henceforth P2017). This paper selects a tomographic sampling of air parcels from a high-resolution chemistry-transport model, meant to simulate what an aircraft mission might measure (e.g., NASA's Atmospheric Tomography Mission: ATom, 2017), and asks if a cohort of six global chemistry models can agree on the reactivity of these parcels. To do this, we develop a new

protocol and set of diagnostics for merging in situ measurements with 3-D model simulations of atmospheric chemistry. We focus here on tropospheric ozone production and loss (P-O3, L-O3) and methane loss (L-CH4), as these two gases are the most important GHGs controlled through tropospheric chemistry. Further, control of $CH_4$ and $O_3$ provides an important pathway for limiting near-term climate change (Shindell et al., 2012). A definition and example of these

reactivities, and how they can be assigned to an air parcel, is found in P2017 and the Supporting Information to this paper.



From the early model-and-measurement assessments that were initiated to support the
stratospheric ozone assessments (NAP, 1983; NASA, 1993), through to the most recent multi-
model evaluations of atmospheric chemistry to be used in upcoming climate assessments
(Collins et al., 2017; Morgenstern et al., 2017; Eyring et al., 2006; Myhre et al., 2017), there is
one truism: the models always produce different results even when they agree upon the
protocols, and intend to do the same simulation. For assessments one seeks common ground to
find a robust result; whereas for science one seeks a cause of disagreement to identify how
models can be improved. This paper focuses on the latter. Given the scale and complexity of
current 3D global chemistry models, potential causes of differences in model-simulated
distributions of chemical tracers is large. The numerical algorithms and parameterizations for
the transport, mixing, and thus dispersion of emissions is clearly one cause (Prather et al., 2008;
Lauritzen et al., 2014; Orbe et al., 2016); while photochemical mechanisms that produce and
destroy species are another (Olson et al., 1997; PhotoComp, 2010).

This paper initiates a new technique for multi-model comparison that uses prescribed initial
chemical composition of air parcels, which we refer to as the modeling data stream. We
presuppose that we can accurately measure or otherwise know atmospheric composition, and
then ask if models calculate the same global chemical budgets for ozone and methane. Our
approach eliminates many of the factors that drive model differences and allows us to focus on
the photochemical reactivities as integrated over a day. Instantaneous reactivities can be inferred
from measurements of reactive chemical species and the radiation field combined with
laboratory cross sections and reaction rate coefficients, e.g., (Olson et al., 2012). Attempts to
follow the chemical evolution of air parcels with aircraft measurements is limited and quasi-
Lagrangian at best (Nault et al., 2016). Even the concept of isolated Lagrangian parcels is
limited, since parcels shear and mix rapidly as they go from a large, chemically coherent air mass
to a heterogeneous mix of smaller features (Batchelor, 1952; Prather and Jaffe, 1990). Yet,
simulating the photochemical changes in $CH_4$ and $O_3$ requires integration over the daily cycle of
photolytic rates, which change greatly and irregularly over the day based on the interaction of the
sun and cloud systems. Unfortunately, there is no known approach to track and measure the 24-
hour net change in ozone or methane for an air parcel in the free troposphere. Here and in
P2017, we approximate the reactivity of an air parcel by running our global chemistry models





with their regular meteorology and chemical modules, but with transport and mixing of tracers
        shut down to keep the grid cells isolated. Effectively, we are able to use the standard full 3D
        model as a collection of box models (i.e., one per grid cell), while incorporating its diurnal cycle
        of photolysis and cloud fields. Such simulations, named the A-runs, are artificial since real air
        parcels constantly move and mix with their environment. Statistical comparison of A-run
        reactivities from the six models with those using the standard 3D versions is examined in P2017,
and shows agreement with some minor biases due to the A-run formulation.

        The participating models and the modeling data stream are described in Section 2. This effort
        was completed before the release of the ATom aircraft data (ATom, 2017) and thus we use a
        1/2°-resolution model to generate the data stream. Section 3 presents and compares the statistics
of P-O3, L-O3, and L-CH4 and J-values from the 14,880 parcels, including 5 different days in
        August to sample variability in cloud systems. Sorted distributions show the models' agreement
        on the most highly reactive parcels. The final discussion in Section 4 considers the role of
        inherent uncertainty in modeling parcel reactivity, of basic differences in the models, and
        whether the new statistics developed here identify and characterize differences in the
photochemical modules. For insight on the most reactive air parcels of the remote troposphere,
        we await a repeat of this work with the ATom data stream.

## 2. Chemistry models and simulations.

The six global chemistry models here are basically the same as those in P2017: Geophysical
        Fluid Dynamics Laboratory (GFDL), Goddard Institute for Space Studies (GISS), Goddard
        Space Flight Center (GSFC), GEOS-Chem (GC), National Center for Atmospheric Research
        (NCAR), and UC Irvine (UCI). For model versions and updates, see Tables 1 and S1.

A model-simulated data stream was prepared from an older version of the UCI model (v72a)
        with higher than usual resolution (T319L60, ~0.55 degrees) and sampled at 00UT 15 August
        2005 at aircraft flight levels along 3 meridians next to 180E. All the model grid cells are used
        with no attempt to follow ATom profiling. This set of 14880 points is similar in number to 10-
        second data from an aircraft mission logging 50 flight hours in the Pacific basin, such as each





seasonal deployment of ATom.  Prescribed species are:  $O_3$, NOx ($=NO+NO_2$), $HNO_3$, $HNO_4$,

PAN (peroxyacetyl nitrate), $RNO_3$ ($CH_3NO_3$ and all alkyl nitrates), HOOH, ROOH ($CH_3OOH$

and smaller contribution from $C_2H_5OOH$), HCHO, $CH_3CHO$ (acetaldehyde), $C_3H_6O$ (acetone),

CO, $CH_4$, $C_2H_6$, alkanes (all $C_3H_8$ and higher), alkenes (all $C_2H_4$ and higher), aromatics

(benzene, toluene, xylene), $C_5H_8$ (isoprene plus terpenes), plus temperature (T) and specific

humidity (q). Zonal mean latitude by pressure plots of $O_3$, CO, HCHO, NOx, PAN and q are

shown in Figure S1.  See Supplement for how these data were implemented in the models and

how two chemistry-climate models were unable to completely overwrite the modeled T&q

values with those from the data stream.

Implications for reactivities are discussed below.  It is difficult, if not impossible, to specify 24-

hour cloud fields, from observations or a model, in a way that all models here could implement

consistently.  Treatment of photolysis rates in uniform cloud layers is still quite different across

models, and fractional overlapping cloud fields are often ignored, e.g., (Prather, 2015).

Likewise, we do not attempt to control the profiles of $O_3$ and aerosol above and below the air

parcels insofar as they impact photolysis.  Hence we diagnose photolysis rates (J values) in

addition to reactivities.

An inherent uncertainty is the day-to-day variability of clouds experienced by each parcel.  Thus

for the single data stream, each model calculates reactivities using the same chemical

initialization but beginning with 5 different days in August: 1st, 6th, 11th, 16th, and 21st.  This 5-

day variance gives us a measure of the uncertainty due to cloud variability, is similar across

models, and thus provides a lower limit on the detection of model-model differences, i.e., a

measure of as-good-as-it-gets in this comparison.

Several uncertainties are not answered with the standard protocol of 5-day runs:  Models ran

with different calendar years and so how do 5-day means vary from year to year? Does the

changing solar declination matter?  Will different restart files (affecting $O_3$ and aerosol profiles)

alter the results? What if the 24h integrations began at midnight rather than noon?  How different

are the CCMs because they use their own T&q for the parcels?  The UCI CTM ran additional

sensitivity calculations to address these questions, see Section 3.5.





| Table 1. Participating models | | | | | | |
|---|---|---|---|---|---|---|
| | model | type | meteorology | T & q | POC | model grid |
| GFDL | AM3 | CCM | NCEP (nudged) | CCM | Arlene Fiore | C180 x L48 |
| GISS | GISS-E2.1 | CCM | daily SSTs, nudged to MERRA | parcel | Lee Murray | 2° x 2.5° x 40L |
| GSFC | GMI-CTM | CTM | MERRA | parcel | Sarah Strode | 1° x 1.25° x 72L |
| GC | GEOS-Chem | CTM | MERRA-2 | parcel | Lee Murray | 2° x 2.5° x 72L |
| NCAR | CAM4-Chem | CCM | MERRA | CCM | Jean-Francois Lamarque | 0.47°x0.625°x52L |
| UCI | UCI-CTM | CTM | ECMWF IFS Cy38r1 | parcel | Michael Prather | T159N80 x L60 |

### 3. Reactivity across the models


The difference in modeled reactivities for each parcel combines variations in cloud fields with basic differences in the chemical models (i.e., chemical mechanisms, numerical methods, photolysis treatment of cloudy and clear sky). The 5-day means reduce the effect of cloud variations but leave the fundamental differences in the photochemical modules, both photolytic

and kinetic reactions. Our comparison looks at the parcel-by-parcel differences including the scatter (rms differences) and average values across the models. To provide a standard for comparisons, we seek a reference case based on several models, and this is easily identified with the rms differences across all model pairs (Table S2). UCI ran 3 different model years to determine a lower-limit rms value, i.e., when the cross-model differences approach this limit, we

can accept that the photochemical modules including clouds cannot be said to be different in this study. For the reactivities (P-O3, L-O3, L-CH4), none of the cross-model pairs reached this lower limit, but certain groupings were consistently close, within a factor of 2 of this limit. For L-O3 and L-CH4, any pair of GSFC-GC-UCI fall within this range, while GFDL, GISS and NCAR are a factor of 5-10 above it. For the two CCMs this is likely caused by their use of

different T&q's, while for GISS it probably lies in the chemical model. For P-O3, only the pair GC-UCI is within a factor of 2, but GFDL-GSFC-GC-UCI form a distinct cluster. The J values, J-O1D (O$_3$+hv=>O$_2$+O($^1$D)) and J-NO2 (NO$_2$+hv=>NO+O), show groupings similar to this cluster, reflecting their common use of Fast-J versions (Wild et al., 2000; Prather, 2015), although this is unlikely to explain their similarity in P-O3.






Based on the average of the 5-day parcel means, we find a cluster of 3 similar models and 3 independent models. We need to find a common reference case against which to plot and statistically evaluate the models. Rather than pick one model, we take the 3-model average, GSFC-GC-UCI, as our reference. This clustering may be due to similar heritage: GSFC and GC are derived from a common tropospheric chemistry module; all 3 models and GISS have a common heritage for photolysis module. In the comparisons below, we will use terms like 'bias' to describe differences with respect to this reference model. Such biases are not meant to be model errors since we do not know the correct answer; they are just model-model differences.

### 3.1. Average profiles

Altitude profiles of reactivities and J values averaged over 24 hours, 5 days in August, and latitude blocks (50S-20S, 20S-20N and 20N-50N) are shown in Figure S2 (6 models, 3 blocks, 18 profiles per panel). As expected for August, the 50S-20S values are very low, while the 20S-20N and 20N-50N ones are equally high. This basic latitude-season pattern holds across all models. The variability across the 5 separate days in the UCI model (Figure S3) is primarily a smooth trend through August reflecting the changing solar declination from 18° to 12°, but instances of highly variable cloud fields occur, even when averaged over 30° in latitude.

For J-O1D, five models (GFDL, GSFC, GC, NCAR, UCI) agree well over all pressures and latitude blocks, but NCAR is, unusually, 10% higher only in the 20S-20N block. J-O1D from GISS is 80% larger than other models for all pressure and latitude blocks, but this does not translate directly or simply into reactivities, where GISS L-O3 is higher (expected) but L-CH4 is lower (unexpected). For J-NO2, model differences are not so great and show largest values at 20N-50N consistent with the longer summer daytime hours. The spread in J-NO2 is partly understandable because of ambiguous choices in interpolating the temperature dependence of recommended $NO_2$ cross sections and quantum yields. This ambiguity does not exist for J-O1D recommended cross section and quantum yields. J-O1D is strongly dependent on the overhead $O_3$ column, and the zonal mean total $O_3$ column from the models is compared with recent satellite measurements in Figure S4. NCAR's $O_3$ column is anomalously lower only in the 20S-20N region and likely explains their higher J-O1D noted above.



Reactivity profiles for the 5 non-GISS models show excellent agreement for P-O3 but noticeable differences for L-CH4 and even larger ones for L-O3 (Figure S2). The altitude profiles are

similar for the 5 models, indicating that the cause of the L-O3 spread is likely related to HOX. The GISS results are anomalous, with much higher P-O3 and an L-O3 vs. L-CH4 relationship that seems counter to known chemistry in which both L-O3 and L-CH4 maximize with the high HOX values in the warmer, wetter, lower troposphere of the tropical Pacific.


*3.2. 14,880 parcels*

We examine the relationship between the 3 reactivities in each model with scatter plots of P-O3 and L-CH4 against L-O3 in Figure 1. Each plot has 14,880 points (5-day parcel means) and is

split by location: 60S-20S & 20N-60N (extra tropics, gray); tropics upper (20S-20N, p < 600 hPa, cyan) and lower (p > 600 hPa, blue). Percentiles (10th, 50th, 90th) in each dimension are plotted as red dash-dot lines, and thus most points in the well correlated L-CH4 vs. L-O3 lie along the 3 quasi-diagonal intersections of red lines. The right-angle separation of high P-O3 and high L-O3 in the tropics reflects the high NOx (P-O3) in this data stream is in the upper

troposphere and the largest L-O3 is from wet environments of the lower troposphere. GFDL has the most compact distribution of parcels and GISS, the most scattered. Four models (GSFC, GC, NCAR, UCI) have remarkably similar patterns in terms of the percentiles and structure, e.g., for L-CH4 vs. L-O3 they show the lower tropics dominating the upper part of the distribution and the extra-tropics, the lowermost points. GFDL has similar percentiles for P-O3 and L-CH4, but a

much smaller spread for L-O3 that explains their compacted scatter plots. GISS is unique with much larger spread in both P-O3 and L-O3 but a compressed distribution in L-CH4. From these scatter plots, we can say that the 4 models are remarkably consistent, that GFDL is similar but should reexamine their L-O3 diagnostic, and that GISS has a 'uniqueness' in its L-O3 vs. L-CH4 relationship as well as large scatter in both P-O3 and L-O3. While consistency does not

guarantee correct implementation of the photochemical model (i.e., rate coefficients, cross sections), uniqueness is something that needs more investigation as it may be an error or may lead to fixes in the 'consistent' models. Scatter plots of J-NO2 and J-O1D vs. L-O3 (Figure S5)





show similar J-value statistics for the 5 non-GISS models, and all models show a similar location of the 3 sets of points (extra-tropics, lower-tropics, upper-tropics) within their own percentiles.


On a parcel-by-parcel basis we compare in Figure 2 the 5-day means from all 6 models against the reference case for the 3 reactivities and 2 J-values. If the models were all alike, they would fall tightly on the 1:1 line (black dashed). In each panel there are 89,280 points with many overlapping. The order of plotting (shown by the legend) is important for visual impression

since the latter points often overlie the earlier ones and the choice of order was based partly on the rms differences, with greatest first and smallest last. Here we can clearly see the type of scatter, the pattern of discrepancies across models, and at what levels of reactivity such discrepancy it occurs. It provides a focus for model development: UCI should reexamine its J-NO2 at the higher values and its P-O3 in the 1-3 ppb/day range; NCAR should examine why it

has so much scatter in L-O3 and L-CH4 (see discussion of T&q later); GFDL has similar scatter (see T&q) but also has a low-bias in L-O3; and GISS has many differences that can be examined. As a cross-model question, are the above-the-line (UCI) and below-the-line (GSFC) differences in P-O3 and L-CH4 related to the same pattern in J-NO2?

A simple summary of these statistics – averages and rms differences relative to the reference case – is given in Table 2. We have selected (bold italics) those entries that seem anomalous as also found in Figure 2. For example, average P-O3 ranges from 0.77 to 0.84 ppb/day for 5 models but is 1.40 ppb/day for GISS. Likewise, average L-O3 ranges from 1.44 to 1.54 ppb/day for 4 models, but is 0.83 for GFDL and 2.25 ppb/day for GISS. The rms differences with respect

to the reference case favors the 3 models that define that case, but also shows that GFDL and NCAR are close to the reference case for P-O3, but farther away for L-O3 and L-CH4 probably caused by their T&q values (see later).

**Table 2.** Average Reactivities and Standard Deviations w.r.t. reference case (average of 3 #'d models).

| Reactivity | P-O3 (ppb/d) | L-O3 (ppb/d) | L-CH4 (ppb/d) | J-NO2 (e-3 /s) | J-O1D (e-5 /s) |
|---|---|---|---|---|---|
| *a) Average Reactivities (5-day averages of 14,880 parcels)* | | | | | |
| Reference Case# | 0.792 | 1.449 | 0.633 | 4.452 | 1.194 |
| GFDL | 0.771 | *0.826* | *0.579* | 4.237 | 1.177 |
| GISS | *1.405* | *2.248* | *0.429* | *5.159* | *2.154* |





| | | | | | |
|---|---|---|---|---|---|
| GSFC# | 0.755 | 1.436 | 0.610 | 4.258 | 1.194 |
| GC# | 0.793 | 1.444 | 0.641 | 4.392 | 1.164 |
| NCAR | 0.839 | 1.541 | 0.666 | 4.475 | 1.305 |
| UCI# | 0.827 | 1.467 | 0.648 | *4.705* | 1.224 |
| *UCI 2015* | *0.833* | *1.474* | *0.651* | *4.725* | *1.227* |
| *UCI 1997* | *0.833* | *1.471* | *0.649* | *4.724* | *1.231* |
| ***b) RMS Differences versus Reference Case, using 5-day means*** | | | | | |
| GFDL | 0.14 | ***0.89*** | ***0.23*** | 0.44 | 0.13 |
| GISS | ***0.84*** | ***1.04*** | ***0.43*** | ***0.93*** | ***1.05*** |
| GSFC# | 0.12 | 0.11 | 0.05 | 0.30 | 0.06 |
| GC# | 0.07 | 0.12 | 0.05 | 0.23 | 0.08 |
| NCAR | 0.16 | ***0.64*** | ***0.23*** | 0.47 | ***0.24*** |
| UCI# | 0.08 | 0.13 | 0.06 | 0.39 | 0.09 |
| *UCI 2015* | *0.10* | *0.17* | *0.08* | *0.48* | *0.11* |
| *UCI 1997* | *0.10* | *0.18* | *0.08* | *0.50* | *0.12* |
| ***c) RMS Differences day-to-day versus 5-day mean of same model*** | | | | | |
| GFDL | 0.09 | 0.26 | 0.12 | 0.36 | 0.08 |
| GISS | ***0.53*** | ***0.41*** | 0.08 | 0.67 | ***0.29*** |
| GSFC | 0.13 | 0.22 | 0.10 | 0.48 | 0.12 |
| GC | 0.09 | 0.19 | 0.08 | 0.43 | 0.10 |
| NCAR | ***0.15*** | ***0.54*** | ***0.21*** | 0.62 | 0.18 |
| UCI | 0.09 | 0.18 | 0.08 | 0.52 | 0.12 |
| *UCI year-to-year* | *0.06* | *0.12* | *0.06* | *0.33* | *0.08* |




Figure 1. Parcel reactivities of (top) P-O3 and (bottom) L-CH4 vs. L-O3 for each of the models. Points are colored by location: 60S-20S & 20N-60N (extra tropics, gray); tropics (20S-20N) upper (p < 600 hPa, cyan) and lower (p > 600 hPa, blue). the 10th, 50th, and 90th percentiles in each dimension are plotted as red dash-dot lines.



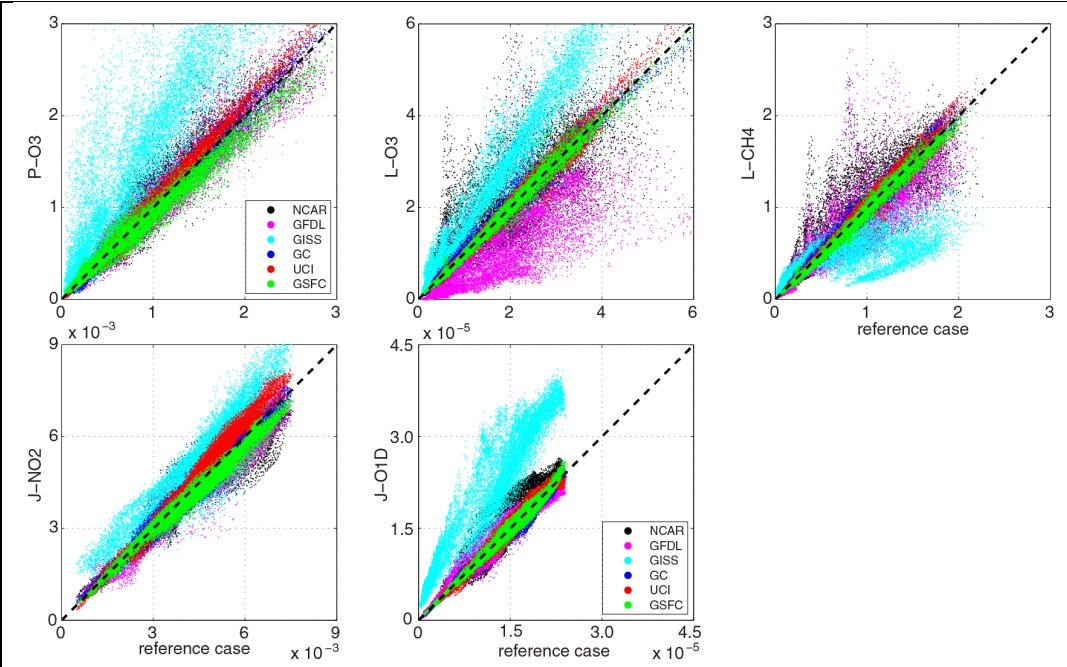

Figure 2. Direct parcel-by-parcel comparison of modeled reactivities (a, P-O3; b, L-O3; c, L-CH4; all ppb/day) and photolysis rates (d, J-NO2; e, J-O1D; all /sec) calculated for the 14,880 simulated air parcels. Each point is an average over the 5 simulated dates in August (8/01, 8/06, 8/11, 8/16, 8/21). The 1:1 line is shown (black dashed) for each plot. The reference values (X axis) are the average of 3 similar models (GSFC, GC, UCI) selected by examining the rms differences across all the models (see text). Note that the model points are plotted successively on top of one another and thus the earlier-plotted models may appear less frequent: in order, NCAR (black), GFDL (magenta), GISS (cyan), GC (blue), UCI (red), GSFC (green).


### 3.3. 5 days versus 5-day mean

The variability of the 5 days in August tells us about the synoptic variability of clouds and possibly $O_3$ columns in each model. The rms difference between the 5 individual days and the 5-

day mean (Table 2) shows that that GISS and NCAR have much larger variability in reactivities, caused by and mirrored by those in J values. These rms differences in J values for GISS and NCAR are surprising. Collectively, we should reexamine this variability in all the models to ascertain its cause. In general, the slopes of the individual versus reference model for reactivities are close to 1 (Table S3) because the slope is determined by the large gradients with latitude and

pressure that most models agree on. In comparing individual days versus 5-day mean, it is encouraging that this slope averages 1±0.04 for all reactivities and models (using each model's 5-



day mean as its reference case, Table S4). Also, the slope decreases from about 1.01 to 0.96 through August as expected with declining photolysis rates in the north.

The rms difference across the 5 days is also a measure of how well the 5-day parcel mean can represent the true chemical model. Assuming that the cloud variability is random, the 5-day means with respect to other models are not really different unless that model-model rms exceeds some fraction of the day-to-day rms of the models involved. Using the UCI test with different model years, we find that the year-to-year rms differences are about 2/3 of the day-to-day rms

over 5-days. Thus, we cannot be sure that the model versus reference case rms values for NCAR are due to the inadequacy of the 5-day mean to represent the mean NCAR chemistry model (Table 2a&c). On the other hand, some other source of model error is likely responsible for the large day-to-day rms.


*3.4 The "hot" air parcels*

Following the "which air matters" theme of P2017, we look at the more reactive air parcels and find out if the models agree on these. For each reactivity, we sort the 5-day parcel means in

increasing order and integrate the cumulative reactivity. The value at 100% (all 14,880 parcels) is equal to the average reactivity of the sample (Table 2a), and this is renormalized to 1 for comparison across models (Figure S6). With sorting, these curves must be monotonic and convex. The steeper the curve, the more important the top reactive parcels are in determining the total. For most all models and reactivities these curves are remarkably similar and fall within the

range seen for 5 different days in the same model (UCI, Figure S7). Focusing on the upper 10%, the outliers are unusual and reactivity specific: for P-O3, one group (GISS-GSFC-NCAR) are less steep than the other (GFDL-GC-UCI), and this splits the reference-case models; for L-O3, GFDL is steeper, consistent with the feature identified earlier in the scatter plots; and for L-CH4, GISS is much less steep although NCAR also deviates slightly in this way from the others. In

this diagnostic, GISS is not such a clear outlier.



From this cumulative reactivity figure, one can see that the top 5% of parcels comprise 15% of the total reactivity, effectively a slope of 3:1. With the exceptions noted, total reactivity for the top 5-10-25-50% of the parcels (Table S5) is similar across models and across days within a

model (Table S6). Focusing on the top 10% of parcels for each reactivity, we plot their latitude-by-pressure distribution for each model in Figure 3. Top P-O3 are in the upper troposphere where NOx was highest in the specified data stream; and top L-O3 and L-CH4 are in the lower troposphere associated with warmer temperatures and higher water vapor, with L-CH4 being at lower altitude than L-O3 (all models except GISS). There is a region of top P-O3 parcels about

40N that extends into the lower troposphere, although the shape varies across models. The vertical pattern of top-10% parcels about 22S clearly varies across models with GISS-GC-UCI not selecting these parcels.

Overlap of these three sets of parcels are quantified as Venn diagrams for each model in Figure

4. Very few top-10 parcels are in the triple-overlap area (1-10%); but when P-O3 parcels coincide with either L-O3 or L-CH4 parcels, they generally lie in this triple-overlap area. The only major exception to this pattern is GISS. In terms of L-O3 and L-CH4 overlap, 4 models are very consistent (76-80%); but GISS is unusually low (49%) and GFDL is unusually high (93%). These patterns help identify distinctly different chemistries in these models that have been

identified with other diagnostics. The Venn overlap diagrams will become more interesting with an observational data stream as they point to the co-occurrence of unusual atmospheric parcels.

At what level do the models agree on the hot, top-10% parcels? We use the reference case defined above and sort each reactivity to identify the top-10%, retain those parcel numbers and

compare across models. Table S7 gives each model's overlap of their top-X% parcels in terms of the percent that also occur in the top-X% reference case. For a range of X, 5%-10%-25%-50%, the overlap increases successively with many models having 90% overlap for the top-50%. The exceptions are GFDL with lower than typical overlap for L-O3 at all top-X% levels, and GISS, with lower overlap for L-CH4. This new diagnostic is helpful in understanding these model

differences because it implies that the L-O3 and L-CH4 differences identified previously are not caused by a systematic offset in all parcels, but rather by a selection of different parcels.





As expected, the three models GSFC-GC-UCI that define the reference case all have about 90%
overlap for the top-10% parcels, and so we do not learn much with this. In terms of linking
models with similar chemistries, probably 80% overlap is a good mark, because we see that the
different UCI years drop off to 85% in L-O3 and L-CH4. Overlap in P-O3 is much easier to
achieve as the few high-NOx parcels drive high P-O3 in all models: at the top-25% parcels, the
P-O3 overlap is about 84% or better for all models.

On a day-to-day basis, we examine the top-10% overlap for GSFC-GC-NCAR-UCI models,
using their own 5-day mean as the reference (Table S8). Cloud variations across the 5 days lead
to overlaps for the top-10% parcels ranging from 78 to 92% at best. NCAR has similar self-
overlaps for P-O3 but only 58 to 72% for L-O3 and L-CH4, because the modeled T&q changes
with each day in August and greatly reduces the overlap of the hot parcels. This further supports
T&q as being important drivers of L-O3 & L-CH4. The use of 5-day calculations with varying
cloud fields is essential in identifying the top reactive parcels.

We plot the modeled reactivity of individual model 5-day mean parcels in ascending order based
on the sorted top-10% parcels in the reference case (Figure S8). Hence the reference case (black
line) is a monotonically increasing curve; while the individual models produce a scattered
distribution of points. As expected, the 3 models defining the reference case have some scatter
but mostly overlap with the reference case. UCI is typically higher and GSFC is lower. For J-
NO2 in these most reactive parcels, UCI is notable higher as is GISS, a result seen in the average
profiles (Figure S2), but it does not affect the reactivities. The mean bias of models relative to
the reference case is also seen in Figure S8 with the offset of the points. The results here are
similar to what has been identified earlier: GISS has unusual offsets for all reactivities and J-
O1D; agreement for P-O3 is much better than for L-O3 and L-CH4; four models show the
upward curve matching the top-1% parcels; for L-O3 and L-CH4, GFDL-NCAR have a flat
scatter of points and miss the upward curve because they reset the q of the data stream. Day-to-
day scatter for the top-10% (defined by the 5-day mean) is tested with the UCI model in Figure
S9. This one-model synoptic cloud variability has similar scatter to that seen for the more
central models (Figure S8) including the rapid increase in L-O3 at the top-1% and the much great
scatter in J-NO2. The year-to-year variability in the top-10% parcels is shown (Figure S10) for




the UCI model with year 2016 as the reference case and years 1997 and 2015 as separate models.
The patterns of scatter here are similar to but less than the day-to-day (Figure S9), again showing

the importance of 5-day averages, and identifying the lower limit of scatter at which this
diagnostic can discern differences in model chemistry.

Note that the J-values have their own top-10% parcels, and that is why J-O1D has better
agreement across models and much smaller scatter than either J-NO2 or the reactivities.  The top

J-O1D values are in the upper troposphere (Figure S2) and less influenced by clouds; whereas
top J-NO2 values occur in cloud fields and even within the clouds.

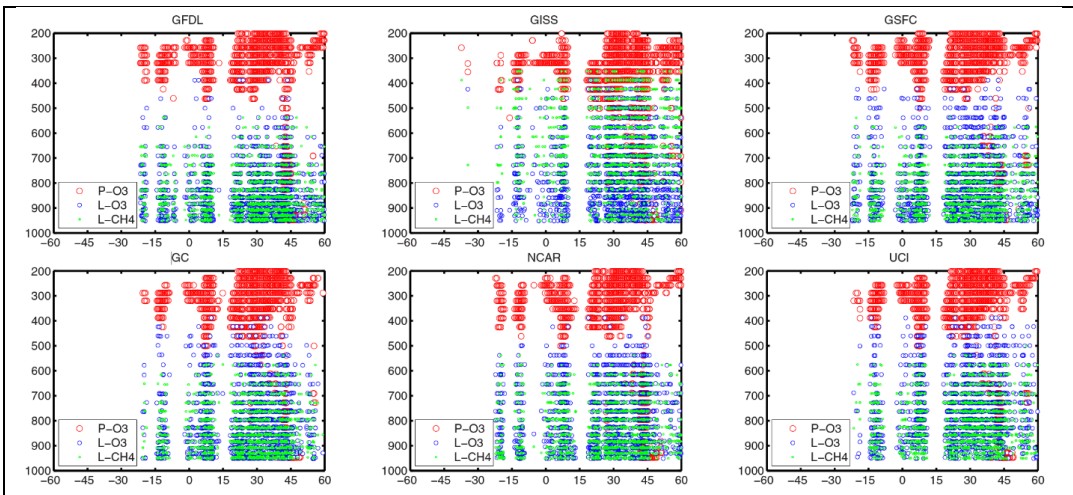

Figure 3.  Latitude (degrees) by pressure (hPa) location of the top 10% of reactive parcels for the 6 models:  P-O3 (red, large circles); L-O3 (blue, medium); L-CH4 (green, small).

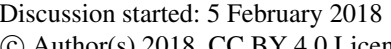



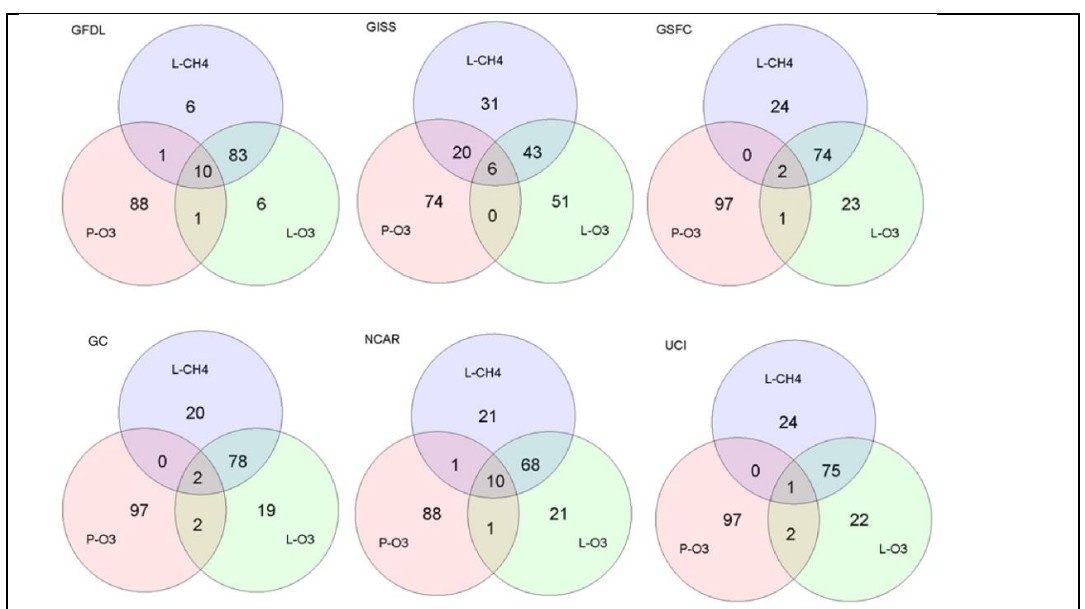

Figure 4. Venn diagrams for each model showing the overlap (%) of the top 10% parcels in each reactivity, using 5-day means for each parcel.


*3.5 Assumptions and uncertainties in the experiment design*

How interannual variability might affect the results is tested with the UCI CTM running the simulated data stream for 5 August days using years 1997 and 2015 meteorology to compare

with year 2016 (see previous Tables, Table S9, and Figure 5). The scatter plots in Figure 5 do not look much different from those for the three models used in the reference case (Figure 2). For the 5-day parcel means, the rms differences across any pairing of the 3 UCI years is about 8-10% of the average reactivity, which surprisingly is about half of that across the 3 models used in the reference case. Using this criterion (<20%) for distinctness, we effectively have only 4

independent distinctly different models here: GFDL, GISS, NCAR and the GSFC-GC-UCI group. The 4 models all differ from one another at the 30-100% level of the UCI year-to-year variations. In terms of the overall average reactivities (Table 2), however, the different years of the UCI model models are almost identical (<1%), while the differences across the 3 reference models are much larger (±5%) and clearly distinguishable.


How the time-of-day of parcels in the data stream might affect reactivity is tested with the UCI model initializing the calculation at midnight instead of noon (see Figure S11, Table S9). In this



study, we chose parcels at 180W and, since the global models begin each day at 0000H UT, the

photochemistry starts at local noon. A measurement data stream, such as from ATom (2017),

will include measurements over a range of longitudes and taken with a wide range of local solar

times. We need to ensure that the protocol here does not depend on when the 24-hour integration

of reactivity is initiated. The UCI model selected one day (8/16) and shifted the local solar time

by 12 hours, thus initiating each parcel at local midnight. In addition, the cloud fields needed to

be rearranged so that the pairing of clouds and solar zenith angles were the same in both cases.

The start-at-midnight version has larger reactivities by at most 1% with no changes in the J-

values as expected for the protocol (e.g., keeping the morning clouds in the morning for both

calculations). The rms differences between the two cases are 2-10 times less than the year-to-

year differences. We conclude that the initiation time produces discernible differences but not at

the level to affect the any of the results here.


Two additional sensitivity tests included running the 5 days in August with a fixed solar

declination (Figure S12) and with different restart file. As shown in the Figure S13 and Table

S9, these two tests change the overall average in the fourth decimal place and have rms

differences <0.01 ppb/day. For these choices, the protocol adopted here is adequate.


The GFDL and NCAR CCMs could not maintain the fixed, data-stream T&q values over the 24-

hour integration, which leads to larger rms differences because reactivities depend on both T and

q. This explains in part why the GFDL and NCAR models in Figure 2 have larger scatter for

reactivities than the other non-GISS models, but similar scatter in J values. This effect may also

contribute to the larger day-to-day rms, for NCAR at least, and is examined more extensively

with the UCI CTM running with the T&q's from both models (Section 3.5).

How overwriting of the data stream's T&q (with a CCM climate) impacts these results is tested

with the UCI CTM re-running a one day (8/16) data stream using T&q's reported out from the

GFDL and NCAR models. The rms reactivity differences for these two models are 2-3 times

larger than those of the reference models (GSFC, GC, UCI, see Table 2); while J-values

differences (much less affected by temperature) are similar.





For the 5x14,880 parcels, the mean values of either GFDL or NCAR T&q's are similar to the
        data stream but their rms differences are large:  about 3.6ºK and 0.4 in $\log_{10}(q)$, see Table S10.
        Both models have similar scatter patterns for T and for q (Figure S14) with a number of parcels
        having $\log_{10}(q)$ more than a factor of 10 different from the stream.  In this sensitivity test, UCI
        CTM ran with just T from GFDL and NCAR, and then with both T&q (4 cases).  The results are
shown in Tables S9 and Figure 6. For T alone, the reactivity differences were at the lower limit
        of detectable model-model differences but, with both T and q, the model showed surprisingly
        large shifts in L-O3 and L-CH4 along with standard deviations 2-10 times larger than the lower
        limit based on different UCI model years.  In fact, the UCI model using GFDL and NCAR T&q
        has about the same rms reactivity differences with respect to the reference case as do the full
models (Compare Tables S9 and 2, noting that Table 2 is a 5-day mean result and not one day).
        Thus, without a model being able to use the specified T&q, we are unable to determine if its
        photochemical module is similar to another model.  Moreover, with climate-varying T&q's the
        modeled reactivities from an observed data stream will also be too noisy for an analysis of the
        top-10% parcels, i.e., which air matters.


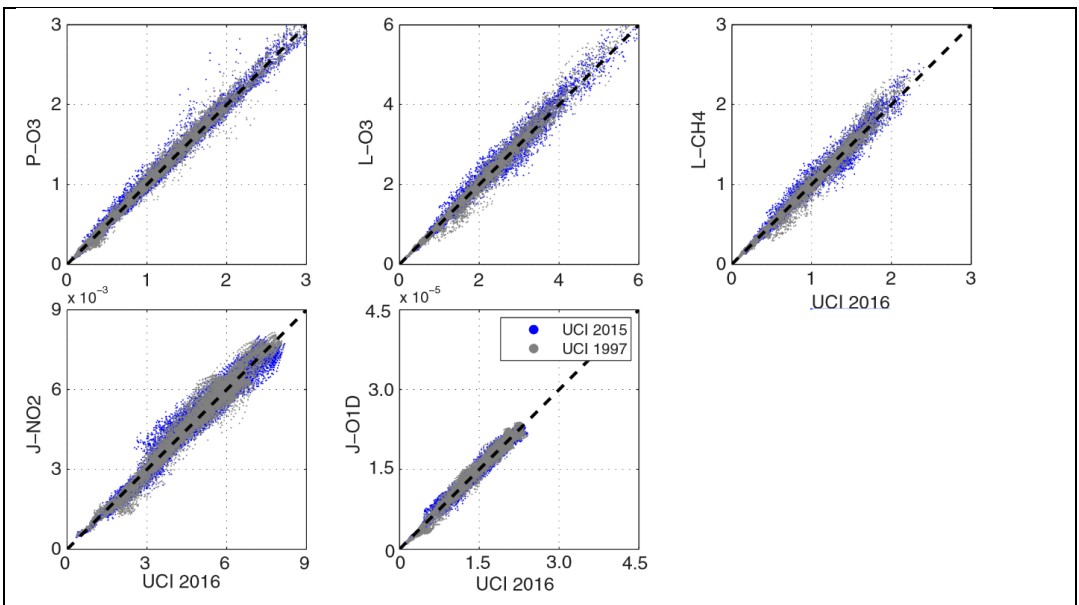

Figure 5.  Scatter plot of reactivities and J-values for 5d-mean air parcels for UCI alternate meteorological years
(2015, 1997) against the standard year 2016.



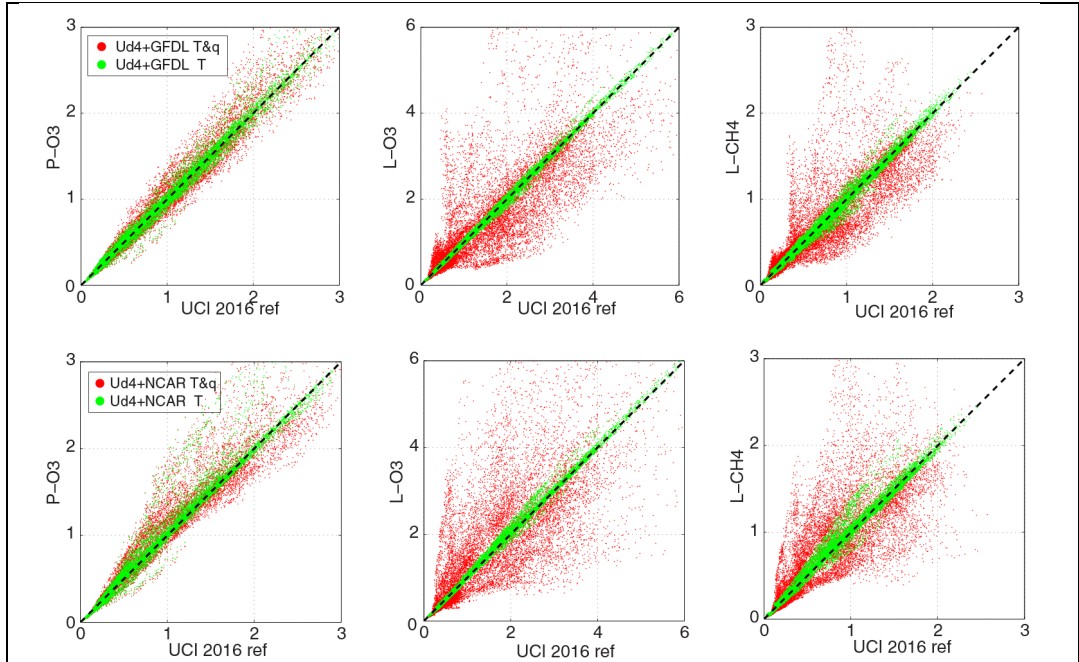

Figure 6. Scatter plot of reactivities of the 14,880 air parcels showing the impact of the GFDL (top) and NCAR (bottom) use of their climate models' T (green) and T&q (red) for a single day (8/16 2016) calculated using the UCI model with both prescribed and these models' T&q values.

*4. Summary discussion*

We develop a new protocol for merging in situ measurements with 3-D model simulations of atmospheric chemistry as calculated by chemistry-transport models through to Earth system models. The goal is to take a time stream of species-rich, high-resolution (100s m), spatially

sparse observations, such as from an aircraft mission (e.g., ATom, 2017), and have the current 3-D global or regional models use that observed data directly to evaluate chemical reactivity in each parcel. With this protocol, we avoid model artifacts in the data stream, such as occur in assimilated data, but must account for the density and bias in sampling. We focus on tropospheric production and loss of the greenhouse gases ozone and methane, but the protocol

can be readily applied to other chemical transformations such as the formation and growth of secondary organic aerosols.





In applying the protocol here to a synthetic data stream, we demonstrate a second major use: detailed diagnostics of model performance, specific to the photochemical modules operating within the global chemistry-climate and chemistry-transport models. Six such models are evaluated here, and their differences and similarities in simulating the chemistry are clearly identified. The protocol specifies the detailed chemical composition of a constrained set of air parcels including temperature and water vapor, embeds these parcels in an appropriate grid cell of each model, turns off processes that mix adjacent grid cells, and integrates the 3-D model for 24 hours (see P2017). The photochemical module is thus dependent only on the chemical mechanism and the diurnal cycle of photolysis rates, which are driven in turn by temperature, water vapor, solar zenith angle, clouds, possibly aerosols and overhead ozone, which are calculated as they would be in each model. Typical 3-D model evaluations cannot separate differences in photochemistry from differences in emissions, transport, scavenging, and even numerical methods, all of which help define the mix of chemical species in each grid cell. This new protocol opens a window focusing specifically on the photochemical modules embedded in 3-D models.

Overall, the models show surprisingly good agreement on calculating the reactivity (P-O3, L-O3, L-CH4) and photolysis rates (J-NO2, J-O1D) in air parcels. We can identify unique features in each model: e.g., UCI's high J-NO2 values; GSFC's lower P-O3 at high reactivity; GISS's inverted results for L-O3 versus L-CH4; GFDL and NCAR's large scatter due to use of model-generated versus parcel-specified water vapor; and large variability in J values for NCAR and GISS. Models with effectively the same chemistry module will appear distinct if they use a different data stream for water vapor. It is impossible to tell if Overall, among the six models, GISS has the most unique features, and GC the least. These anomalous features can really only be explained by the model developers who understand the coding, yet these diagnostics point to a focus for the analysis of individual models. Being a standout in any diagnostic, does not necessarily imply that uniqueness is an error, but it should encourage self-evaluation to determine if that unique feature is intentional and can be shown to be a more accurate simulation.

Cloud variations on synoptic scales are primary sources of noise in this study. These are difficult to standardize from either model or observation given the wide range of methods for treating



cloud scattering and overlap. Cloud-driven changes in reactivity are clear in comparisons across

models and also within the same model. Use of a single day for comparison is inadequate. This
protocol selects 5 days across the month to sample cloud fields, and this provides a stable
average for identifying model-model difference. The protocol also makes several simplifying
assumptions that may affect results: the solar declination over the month is fixed at the mid-
month value; and the 24-hour integration is always started globally at the same universal time,

meaning at different local solar times across the longitudes. These issues were tested with a
single model and found to be unimportant compared with the synoptic variability in clouds and
other model-model differences.

Using day-to-day and year-to-year variability in a single model, we can define a lower limit to

the differences, which is essentially the noise in this protocol, such that models are not
distinguishably different. For the most part, we find that the GSFC, GC and UCI models fall into
this indistinguishable-from-one-another class because their differences are within a factor of 2 of
the estimated noise level. This grouping may be explained in part by the common heritage of
GSFC and GC's tropospheric chemical model, but UCI's chemical mechanism is completely

different and much abbreviated. All other model pairings show much larger differences.

All models agree that the more highly reactive parcels dominate the chemistry; for example, the
hottest 10% of parcels control 25-30% of the total reactivities. Unfortunately, they do not agree
on which parcels comprise the top 10%. This diagnostic will become more acute as we move

from the smoothed synthetic data stream derived from model output (50 x 50 x 1 km averages) to
the high variability of in situ ATom observations (2 x 2 x ~0.1 km averages).

Based on our experience comparing models that differ largely by temperature and water vapor,
we conclude that water vapor differences in CCM simulations of past and future atmospheres

may be a major cause of the changes in $O_3$ and $CH_4$ and may lead to different chemistry-climate
feedbacks across the models.





***Acknowledgments***.  This work was supported by NASA funding of the EVS2 Atmospheric

Tomography (ATom) mission through a range of specific funding mechanisms. We thank

Jingqiu Mao and Larry Horowitz for assistance with GFDL AM3, and Drew Shindell for

assistance with GISS model 2E.

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
