# Peer review of "How well can global chemistry models calculate the reactivity of short-lived greenhouse gases in the remote troposphere, knowing the chemical composition"

_Atmospheric Measurement Techniques, 2017_

## Referee Comment (RC1) · Anonymous Referee #1 · 23 Mar 2018

Prather et al report a new modelling protocol for merging in situ measurements of reactive chemical species into 3D models that simulate their chemistry. The goal of this work is to enable new insight into how the chemistry schemes and processes in these 3D models affect the results of simulations performed with them by isolating (switching of) meteorological mixing, emissions and deposition. This protocol is tested with a synthetic (model derived) data stream in the current work but could be extended to a real observational data set when such a data set becomes available (i.e. from the ATom mission dataset).

[Figure]

This is a very original idea – albeit an extension, in some way, of years of previous work using single box models. It is a neat idea and as a global modeller not involved in this work I will be working out how to apply this protocol to my model to test my model's results! I would recommend publication in AMT following the adoption of the following minor points.

My major annoyance with this paper was the constant reference in the main paper to figures and tables in the supporting information (SI). I would recommend large parts of the SI be moved into the main manuscript.

**General comment**:

- Many of the figures are low resolution. Please make sure that the final images are higher quality scalable vector images. In addition, many of the figures have no units on the axes. Please check them and add units where appropriate.

- Please also ensure consistent use of colours for models. There are many figures and it makes it easier to keep track if colour is used consistently. For example in Figure 2 NCAR is black but Figure S2 NCAR is magenta.

**Specific comments**:
Page 1, line 21: "the data" I think should be "these data".
Page 2, line 36: Are the regions being referred to spatial or chemical or both?
Page 2, line 60: Please add the information referenced in the SI on **Reactivities** to the main document.
Page 3, line 72: How large is large? Can this be quantified?
Page 3, line 75: This is a minor point but as I understand PhotoComp 2010 mainly

assessed stratospheric chemistry? The work in this paper focuses on tropospheric reactivities. What relevance is this reference to the present work?

Page 4, line 118: I think this refers to Tables S1a and S1b not S1?

Page 5, line 131: I think it's important that the information referred to here from the SI be put into the main paper. This is a technique paper and it is painful to have to refer to the SI to actually find out about the technique in question.

Page 6, line 168: Define RMS.

Page 6, line 168: It's not clear why 3 different model years would be capable of defining a lower-limit for the RMS? Why not 2, 5 or 10 years?

Page 6, line 168: I'd suggest adding "(in blue in Table S2)" after "lower-limit rms". Page 7, line 206: Are there any references for this statement of "ambiguous choices"? Will other modelling groups know what you mean here. I'm afraid I don't.

Page 12, line 280: Correct the duplicate "that" typo.

Page 13, line 315: From Figure S6 it seems to me that GISS is an outlier for L-CH4. Can you clarify why you state it is not?

Page 15, line 376: Typo – "great" should be "greater".

Page 16, line 378: I would suggest adding "(solid line)" after "reference case". Page 16, line 386: Looking at Figure S2 I don't see that the largest J-NO2 values are within clouds. It looks to me that J-NO2 follows J-O1D and increases to a maximum in the upper troposphere.

Page 16, Figure 3: Please add x and y axis information to the plots.

Page 17, Figure 4: I like the figure! But, the colours may be difficult for a colour blind person to interpret. I have had a go at uploading and examining the figure here (http://www.color-blindness.com/coblis-color-blindness-simulator/). As far as I can see this may be OK, but I would advise as a general note to consider the use of colour more.

Page 17, line 398: Why is this surprising?

Page 17, line 403: Remove typo "models".

Page 18, line 412: Please clarify that 8/16 is 16th August. As a non-American English

user I was a bit confused but guessed this is what you meant?

Page 18, line 419: Are these small differences likely in all environments? Is this protocol only valid for the remote Pacific? What levels of NO3 and N2O are present and how would a more vigorous night-time chemistry effect these conclusions?

Page 18, line 435: Delete "out".

Page 18, line 437: I think "J-values" should be "J-value".

Page 19, line 440: I think removing "5x14,880" and replacing with "5 days that we consider 14,880 parcels per day," would make this a bit clearer for the reader.

Page 20, Figure 6: What is "Ud4"? in the legend? Delete "use of their" from the caption.

Page 21, line 495: Decapitalise "Overall".

SM-1: There is a typo in the units for P-O3.

SM-2: It is not clear how you have adopted the partitioning of collective species? For example, in my model I only have 3 NMVOC. My ATom or synthetic data set has 12 NMVOC. What should I do? I think it would be nice if the UCI data were made available as a data source for other modellers to use and then they can compare their results to this paper.

SM-4, Table S1b: Can the versions of codes to calculate photolysis rates be added? Similarly, can references for the chemistry schemes or the papers that describe the schemes be added. I'm sure GFDL have updated some of the rates that were in the original MOZART-2 scheme since it was developed. ASAD, as I understand is not so much a chemical mechanism but a software for integrating chemistry. What is meant by "Wild, FRSGC"? Is this a reference?

Table S6, S7 and S8: What is "R"?

Figure S1: There is a growing consensus that the rainbow/jet colour scale should not be used for quantitative inference (Hawkins et al., 2015). I would consider switching the colour scale used to something like the viridis colour scale.

Figure S1-S9: See general comment about consistent use of colour. NB I can not see a difference in colour between GFDL and GISS in the legend of Figure S8.

Figure S11: What is "Ud4"?

Figure S12: Are the different dates important here? If not a hex-bin plot would be much nicer to help the reader see the relationships between the data.

**References**:

Hawkins, Ed. "Graphics: Scrap rainbow colour scales." Nature 519, no. 7543 (2015): 291.
* * *

---

## Referee Comment (RC2) · Anonymous Referee #2 · 9 Apr 2018

The manuscript presents the development of a new protocol for evaluating photochemistry and reactivity of air masses comparing different models with in situ measurements of atmospheric composition. The authors use synthetic in situ measurements to demonstrate the feasibility and application of their protocol. The topic of the manuscript is very timely given the wide availability of atmospheric chemistry models and observations and the growing interest in such models and observations for air quality and climate applications. I recommend publication subject to the following comments:

General comments:

[Figure]

The manuscript is sometimes very heavy in terms of the text and some adjustments to better illustrate the points would be beneficial. Some of the figures in the supplement may be better suited in the main manuscript, particularly in Section 2 describing the overall comparison of the models in terms of the key species and the profiles of their reactivity. I suggest moving some of these to the main manuscript.

In the summary the authors refer to the ability of 3-D models to separate the effects of photochemistry and emissions, for example, and a brief comment on models that can do this, and the advantage of their protocol over such models could be beneficial to the reader.

It is clear the use of this protocol for identifying and evaluating inter-model differences is very useful and some bullet points on the main highlights of the findings in this regard could be helpful.

Specific comments:

Page 8, line 218: 'x' should be subscript in HOx

Page 13, lines 303-304: should "and find out" be "to find out"?

Page 13, line 309: the beginning of the sentence "For most all models" doesn't make a lot of sense, please clarify

Page 15, line 367: should "notable" be "notably"?

Page 18, line 412: it isn't clear what is meant in describing the selected day - does 8/16 refer to a generic or specific day in August 2016? This is also referred to later in the manuscript.
* * *

---

## Author Comment (AC1) · 11 Apr 2018

Prather et al report a new modelling protocol for merging in situ measurements of reactive chemical species into 3D models that simulate their chemistry. The goal of this work is to enable new insight into how the chemistry schemes and processes in these 3D models affect the results of simulations performed with them by isolating (switching of) meteorological mixing, emissions and deposition. This protocol is tested with a synthetic (model derived) data stream in the current work but could be extended to a real observational data set when such a data set becomes available (i.e. from the ATom mission dataset).

This is a very original idea – albeit an extension, in some way, of years of previous work using single box models. It is a neat idea and as a global modeller not involved in this work I will be working out how to apply this protocol to my model to test my model's results! I would recommend publication in AMT following the adoption of the following minor points.

My major annoyance with this paper was the constant reference in the main paper to figures and tables in the supporting information (SI). I would recommend large parts of the SI be moved into the main manuscript.
*OK, have moved many of these into the main paper. In particular, we have brought all the 6-model plots into the main paper, but left the one-model sensitivity test figures in the Supplement.*

General comment:
• Many of the figures are low resolution. Please make sure that the final images are higher quality scalable vector images. In addition, many of the figures have no units on the axes. Please check them and add units where appropriate.
*The .png figures in the .doc file look sharp but my conversions to .pdf made them fuzzy – sorry, should have looked more closely. In the final, all figures are fully resolved .eps*

• Please also ensure consistent use of colours for models. There are many figures and it makes it easier to keep track if colour is used consistently. For example in Figure 2 NCAR is black but Figure S2 NCAR is magenta.
*Agreed. I am still learning about how to use colors with matlab and will implement better, consistent colors!*

Specific comments:
Page 1, line 21: "the data" I think should be "these data".
*done, thanks.*

Page 2, line 36: Are the regions being referred to spatial or chemical or both?
*done, 'spatial'.*

Page 2, line 60: Please add the information referenced in the SI on Reactivities to the

main document.

*done.*

Page 3, line 72: How large is large? Can this be quantified?

*Miswrote here, the subject is 'causes' and the predicate is now 'are many.'*

Page 3, line 75: This is a minor point but as I understand PhotoComp 2010 mainly assessed stratospheric chemistry? The work in this paper focuses on tropospheric reactivities. What relevance is this reference to the present work?

*Actually PhotoComp performed a range of photolysis, including J-NO2 with tropospheric cloud layers.*

Page 4, line 118: I think this refers to Tables S1a and S1b not S1?

*done, thanks.*

Page 5, line 131: I think it's important that the information referred to here from the SI be put into the main paper. This is a technique paper and it is painful to have to refer to the SI to actually find out about the technique in question.

*OK, OK. Have expanded:*
*"The implementation of this data stream of reactive species is model dependent.  All models begin with their own 3D initialization data set that is used to restart a model simulation beginning on August 16.  The specified air-parcel NOx, for example, will be initialized as separate NO and NO2 abundances by scaling the model's restart values for NO and NO2 to match the specified parcel NOx.  Similarly, a single value for aromatics will be partitioned over benzene, toluene, and xylene by models that resolve these species in accord with the restart values.  The models place each parcel (i.e., overwrite the restart values) in the grid cell containing the latitude, longitude, and pressure specified for that parcel.  If that preferred grid cell is already occupied with an air parcel, then an alternate adjacent grid cell is selected.  It is recommended that alternate cells be shifted to minimize the change in photolytic environment (e.g., shift by longitude but maintain surface albedo and atmospheric mass).  Two chemistry-climate models (GFDL, NCAR) were unable to completely overwrite the modeled T&q values with data stream values (see sensitivity tests below).  See also Supplement for additional details."*

Page 6, line 168: Define RMS.

*done, thanks.*

Page 6, line 168: It's not clear why 3 different model years would be capable of defining a lower-limit for the RMS? Why not 2, 5 or 10 years?

*The choice of 3 seemed reasonable to test 3 different pairings. The sentence have been revised for clarity: "UCI ran 3 different model years to estimate the rms value caused by interannual variability (blue in Table S2), i.e., when the cross-model differences approach this value, we can accept that the photochemical modules including clouds cannot be said to be different in this study."*

Page 6, line 168: I'd suggest adding "(in blue in Table S2)" after "lower-limit rms".

*done, but part of revised sentence above.*

Page 7, line 206: Are there any references for this statement of "ambiguous choices"? Will other modelling groups know what you mean here. I'm afraid I don't.

*Sorry, I thought this was well known, but will be explicit and add: "(i.e., the absorption cross sections are given at 220K and 294K; the quantum yields, at 248K and 298K; and the choice of whether to interpolate linearly or logarithmically, or whether to extrapolate or not, affects J-NO2, especially in the upper troposphere). "*

Page 12, line 280: Correct the duplicate "that" typo.

*done.*

Page 13, line 315: From Figure S6 it seems to me that GISS is an outlier for L-CH4. Can you clarify why you state it is not?

*Sorry, this appears to have been written with an earlier set of figures, it requires a major re-write: "Focusing on the upper 10%, the outliers are unusual and reactivity specific: for L-O3, GFDL is much steeper that the other models, consistent with the feature identified earlier in the scatter plots; and for L-CH4, GISS is much shallower. Surprisingly, with this diagnostic GISS is not an obvious outlier for P-O3 and L-O3 as seen in previous comparisons."*

Page 15, line 376: Typo – "great" should be "greater".

*done, thanks.*

Page 16, line 378: I would suggest adding "(solid line)" after "reference case".

*done, thanks.*

Page 16, line 386: Looking at Figure S2 I don't see that the largest J-NO2 values are within clouds. It looks to me that J-NO2 follows J-O1D and increases to a maximum in the upper troposphere.

*Correct. Fig S2, because it is averages, and does not show the cloud levels. I will make the statement because it does explain the patterns and does follow from our (not shown) studies.*

Page 16, Figure 3: Please add x and y axis information to the plots.

*done, thanks.*

Page 17, Figure 4: I like the figure! But, the colours may be difficult for a colour blind person to interpret. I have had a go at uploading and examining the figure here (http://www.color-blindness.com/coblis-color-blindness-simulator/). As far as I can see this may be OK, but I would advise as a general note to consider the use of colour more.

*Good point, but not really necessary as the colors are the same for all the Venn diagrams and only used to guide the eye. I will see how easy it is to redo the colors.*

Page 17, line 398: Why is this surprising?

*Good point, dropped.*

Page 17, line 403: Remove typo "models".
    *done, thanks.*

Page 18, line 412: Please clarify that 8/16 is 16th August. As a non-American English user I was a bit confused but guessed this is what you meant?
    *OK, put "August 16"*

Page 18, line 419: Are these small differences likely in all environments? Is this protocol only valid for the remote Pacific? What levels of NO3 and N2O are present and how would a more vigorous night-time chemistry effect these conclusions?
    *I really do not want to get into a lot of 'what ifs' here. A caveat sentence has been added.*

Page 18, line 435: Delete "out".
    *done, thanks.*

Page 18, line 437: I think "J-values" should be "J-value".
    *done, thanks.*

Page 19, line 440: I think removing "5x14,880" and replacing with "5 days that we consider 14,880 parcels per day," would make this a bit clearer for the reader.
    *Yes, fixed another way.*
Page 20, Figure 6: What is "Ud4"? in the legend? Delete "use of their" from the caption.
    *OK, need to rewrite the caption for clarity.  Ud4 is UCI running day 4 (Aug 16) with the GFDL or NCAR T's and T&q's.*

Page 21, line 495: Decapitalise "Overall".
    *done, thanks.*

SM-1: There is a typo in the units for P-O3.
    *done, thanks.*

SM-2: It is not clear how you have adopted the partitioning of collective species?
For example, in my model I only have 3 NMVOC. My ATom or synthetic data set has 12 NMVOC. What should I do?
    *Good point, I have added: " The algorithm for dealing with missing species or an over-specified class of species is truly model dependent.  For example, the UCI model has a simple approximation and single class for all aromatics and consolidates emissions of benzene, toluene, and xylene into 'aromatics' that react as benzene.  The NCAR model includes all three species explicitly, and thus they will take the mole fraction of 'aromatics' and partition it into benzene, toluene, and xylene, scaled to their values in the grid cell that is being overwritten with the UCI data stream. "*

I think it would be nice if the UCI data were made available as a data source for other modellers to use and then they can compare their results to this paper.

*Humble apologies, yes, of course we will publish the data stream used here with the SM.*

SM-4, Table S1b: Can the versions of codes to calculate photolysis rates be added? Similarly, can references for the chemistry schemes or the papers that describe the schemes be added. I'm sure GFDL have updated some of the rates that were in the original MOZART-2 scheme since it was developed. ASAD, as I understand is not so much a chemical mechanism but a software for integrating chemistry. What is meant by "Wild, FRSGC"? Is this a reference?

*I will try to extract some more details, but a full documentation of the chemistry is beyond this work. The UCI notes-to-self that you noted have been replaced with a proper reference.*

Table S6, S7 and S8: What is "R"?

*Sorry, that was lazy, have spelled out "reactivity"*

Figure S1: There is a growing consensus that the rainbow/jet colour scale should not be used for quantitative inference (Hawkins et al., 2015). I would consider switching the colour scale used to something like the viridis colour scale.

*A reasonable point, but this figure is not primary, and will leave as is for now. Nevertheless, I am reworking the colors and legends for the other key figures.*

Figure S1-S9: See general comment about consistent use of colour. NB I can not see a difference in colour between GFDL and GISS in the legend of Figure S8.

*Yes, see above. We are moving S2, S6, and S9 into the main text.*

Figure S11: What is "Ud4"?

*Yes, revised caption and figure. No need to specifically call out UCI day 4 (8/16).*

Figure S12: Are the different dates important here? If not a hex-bin plot would be much nicer to help the reader see the relationships between the data.

*Yes, normally the dates 8/01 – 8/21 see a drop in solar declination as the sun moves south. We needed to check that this caused very small spread. The essence of most of these 'non-result' plots is that our assumptions are OK.*

References:
Hawkins, Ed. "Graphics: Scrap rainbow colour scales." Nature 519, no. 7543 (2015): 291.

Anonymous Referee #2

The manuscript presents the development of a new protocol for evaluating photochemistry and reactivity of air masses comparing different models with in situ measurements of atmospheric composition. The authors use synthetic in situ measurements to demonstrate the feasibility and application of their protocol. The topic of the manuscript is very timely given the wide availability of atmospheric chemistry models and observations and the growing interest in such models and observations for air quality and climate applications. I recommend publication subject to the following comments:

General comments:

The manuscript is sometimes very heavy in terms of the text and some adjustments to better illustrate the points would be beneficial. Some of the figures in the supplement may be better suited in the main manuscript, particularly in Section 2 describing the overall comparison of the models in terms of the key species and the profiles of their reactivity. I suggest moving some of these to the main manuscript.

*Yes. Some of the text adjustments have been made in our re-reading and also in response to RC1's specific comments. We have moved all of the critical model-model comparisons to the main text (the covariance Table S2, Figures S2, S6, S8) and left the sensitivity analysis (with the mostly, hoped-for, null results) in the Supplement.*

In the summary the authors refer to the ability of 3-D models to separate the effects of photochemistry and emissions, for example, and a brief comment on models that can do this, and the advantage of their protocol over such models could be beneficial to the reader.

*We are not sure quite what is wanted here, but that topic (at the end of paragraph #2 in the Summary) has been split out, better explained.*

It is clear the use of this protocol for identifying and evaluating inter-model differences is very useful and some bullet points on the main highlights of the findings in this regard could be helpful.

*Hmm. We have added a short paragraph at the end: "This new protocol for multi-model evaluations helps identify and provide insights into inter-model differences, as well as providing a direct link with measurements made at a much finer scale than the model.*

Specific comments:
Page 8, line 218: 'x' should be subscript in HOx

*done, thanks.*

Page 13, lines 303-304: should "and find out" be "to find out"?

*Yes, this is better- done.*

Page 13, line 309: the beginning of the sentence "For most all models" doesn't make a lot of sense, please clarify

*Yes, identified by both reviewers – this has been changed to "For most models, these reactivity curves…*

Page 15, line 367: should "notable" be "notably"?

*Thanks, done.*

Page 18, line 412: it isn't clear what is meant in describing the selected day – does 8/16 refer to a generic or specific day in August 2016? This is also referred to later in the manuscript.

*Our original notation did not work well when included as text.  RC1 also had trouble with this.  We have explicitly stated the date in the UCI model and dropped references to day 4 or 'd4' (which happened to be the 4$^{th}$ of the 5-day sequence August 1, 6, 11, 16, 21 that were sued to assess cloud variability. This should be throughout the ms.*

---

## Author Comment (AC2) · 11 Apr 2018

Please see response to RC#1 and the .pdf file attached to that response.